# Changes in Ultrasound Measurements of the Ulnar Nerve at Different Elbow Joint Positions in Patients with Cubital Tunnel Syndrome

**DOI:** 10.3390/s22218354

**Published:** 2022-10-31

**Authors:** Tomasz Wolny, César Fernández-de-las-Peñas, Arkadiusz Granek, Paweł Linek

**Affiliations:** 1Musculoskeletal Elastography and Ultrasonography Laboratory, Institute of Physiotherapy and Health Sciences, The Jerzy Kukuczka Academy of Physical Education, Mikołowska 72A, 40-065 Katowice, Poland; 2Department of Physical Therapy, Occupational Therapy, Rehabilitation and Physical Medicine, Universidad Rey Juan Carlos, 28922 Madrid, Spain; 3Hospital of the Ministry of Interior and Administration, 25-316 Kielce, Poland

**Keywords:** ultrasound imaging, shear-wave elastography, cubital tunnel syndrome, ultrasound, entrapment neuropathy, rehabilitative ultrasound imaging

## Abstract

Ultrasound imaging (US) is increasingly being used in the diagnosis of entrapment neuropathies. The aim of the current study was to evaluate changes in stiffness (shear modulus), cross-sectional area (CSA), and trace length (TRACE) of the ulnar nerve in patients with cubital tunnel syndrome (CuTS), with shear wave elastography (SWE). A total of 31 patients with CuTS were included. CSA, shear modulus, and TRACE examinations were performed in the SWE mode in four positions of the elbow: full extension, 45° flexion, 90° flexion, and maximum flexion. There were significant side-to-side differences in the ulnar nerve elasticity value at 45°, 90°, and maximal elbow flexion (all, *p* < 0.001) but not at elbow extension (*p* = 0.36). There were significant side-to-side differences in the ulnar nerve CSA value at each elbow position (all, *p* < 0.001). There were significant side-to-side differences in the ulnar nerve trace value at each elbow position (all, *p* < 0.001). The symptomatic ulnar nerve in patients with CuTS exhibited greater stiffness (shear modulus), CSA, and TRACE values, compared with the asymptomatic side. US examinations (shear modulus, CSA, and TRACE evaluation) of the ulnar nerve can be helpful in supporting and supplementing the diagnosis in patients with CuTS.

## 1. Introduction

Cubital tunnel syndrome (CuTS) is the second most common entrapment neuropathy of the upper extremity after carpal tunnel syndrome and is the most common neuropathy of the ulnar nerve [1,2]. The prevalence of CuTS is estimated between 1.8% and 5.9% [3]. In the early stages of CuTS, sensory symptoms (paresthesia) are predominant, which occur paroxysmally, and, in more advanced stages, there is the weakness of the intrinsic muscles of the hand and potential atrophy [4]. The disease is progressive, and the gradual impairment of hand function can adversely affect many activities of daily living, professional work, and a sense of the overall quality of health.

CuTS is difficult to diagnose due to the large spectrum of clinical symptoms and the limitations of electrophysiological testing [5,6,7]. Some researchers have advised about the lack of a gold standard in the diagnosis of CuTS [8], while others have identified nerve conduction as the gold standard for diagnosing CuTS [5,9]. However, researchers point to low sensitivity, the risk of false-negative results, and the invasiveness of nerve conduction [10,11]. In clinical practice, diagnosis is based on a combined analysis of the patient’s history, clinical symptoms, physical examination, and electrophysiological testing [12]. In the diagnosis of CuTS, the relevance of ultrasound imaging (US) is increasingly indicated [12,13,14].

US is a promising tool in the diagnosis of peripheral neuropathies [15], including CuTS [7,14,16]. It can allow an increase in the accuracy of CuTS diagnosis [7,14,16,17]. The most common US assessment is the measurement of the cross-sectional area (CSA) of the nerve, as the nerve tends to flatten at the site of compression and swell proximally to the compression site [7,14]. Some authors use the flattening index in the cubital tunnel by calculating it as the ratio of transverse to anteroposterior dimensions [7] and also evaluate the echogenicity of the ulnar nerve [14]. The first reports have been published recently on the potential usefulness of using shear-wave elastography (SWE) in the diagnosis of CuTS, with greater ulnar nerve stiffness in the cubital tunnel seen in patients with CuTS [18,19]. In both studies, the authors emphasized that SWE is a reliable and simple quantitative method to aid in the diagnosis of CuTS [18,19]. Thus, conducting further research on the role of US (especially SWE) in the diagnosis of CuTS seems highly desirable.

To the best of the authors’ knowledge, no studies have evaluated US measurements (SWE, CSA, TRACE) of the ulnar nerve in the cubital tunnel at different angles of elbow joint flexion in patients with diagnosed CuTS. It can be speculated that the reduced lumen of the cubital tunnel and the accompanying compression of the tissues surrounding the nerve and its increasing stretching [20,21] for different angular positions of the elbow joint may affect the SWE, CSA, and TRACE of the ulnar nerve. This, in turn, may allow for a better understanding of US lesions of the ulnar nerve in CuTS patients and influence the further development of the diagnostic role of US. Therefore, the aim of the current study was to compare ultrasonographic changes (shear modulus, CSA, and TRACE) in the cubital tunnel of the symptomatic extremity (CuTS) with the asymptomatic extremity in a sample of individuals with CuTS.

## 2. Methods

### 2.1. Study Design

This was an observational study with repeated measurements, including subjects with clinically and electrophysiologically diagnosed CuTS. The examinations were conducted in a medical outpatient clinic and Musculoskeletal Elastography and Ultrasonography Laboratory (The Jerzy Kukuczka Academy of Physical Education). All US examinations were performed by a therapist with more than 10 years of experience in peripheral nerve ultrasound examinations. The US examinations evaluated CSA, shear modulus, and TRACE of the ulnar nerve on the symptomatic and asymptomatic extremities. The assessor was blinded to the side of the symptoms. Participants in the study were informed not to give any information about their health status to the assessor taking the measurements. All participants were informed about the study. Written informed consent was obtained from all participants. All study procedures were performed according to the Declaration of Helsinki of 1975 and revised in 1983. The study was approved by the Bioethics Committee for Scientific Research of the Jerzy Kukuczka Academy of Physical Education in Katowice (No. 8/2019).

### 2.2. Participants

The study included consecutive persons who presented to the outpatient clinic for physiotherapy management and had had CuTS diagnosed clinically and electrophysiologically by a physician. The criteria for inclusion of CuTS patients in the study were clinical symptoms of peripheral neuropathy of the ulnar nerve (pain, numbness, tingling, and sensory disturbances in the innervation of the ulnar nerve) and below-normal nerve conduction (motor conduction velocity < 49.3 m/s), unilateral CuTS, and absence of CuTS symptoms in the other upper extremity (nerve conduction examinations of the asymptomatic extremity were not performed for ethical reasons). Exclusion criteria consisted of lack of consent to participate in the study, bilateral CuTS, previous upper extremity surgery, use of steroid pharmacotherapy and non-steroidal anti-inflammatory drugs within the past 6 months, radiologically and electrophysiologically confirmed cervical radiculopathy, carpal tunnel syndrome, diabetes, and rheumatic diseases.

### 2.3. Ultrasound Measurements

All US measurements were performed using an Aixplorer ultrasound scanner (Product Version: 12.2.0; Sofware Version: 12.2.0.808; Supersonic Imagine, Aix-en-Provence, France) using a linear transducer array (2–10 MHz; SuperLinear 10-2, Vermon, Tours, France). Before the US examination, points (the olecranon and the medial humeral epicondyle) were marked on the skin with a marker. A point on the medial humeral epicondyle also served as an application site for the axis of rotation of the goniometer, which allowed us to control the degree of flexion of the elbow joint (Figure 1). All participants were examined in a side-lying position with the lower limbs flexed at the hip and knee joints and with a pillow under the head. The upper limb lying above was placed along the torso. The lower-lying limb was examined, and positioning on the side allowed for easy measurement and control of the degree of flexion in the elbow joint with the goniometer and a better view of the cubital tunnel.

CSA, shear modulus, and TRACE examinations were performed in the SWE mode in four positions of the elbow joint (full extension, 45° flexion, 90° flexion, and maximum flexion) with a transverse probe position (Figure 2). The CSA is the cross-section of the ulnar nerve in the ulnar tunnel measured in square millimeters (mm^2^). The share modulus evaluated ulnar nerve stiffness in kilopascals (kPa). The TRACE, in turn, is the circumference of the ulnar nerve over the inner hypoechoic border of the ulnar nerve measured in millimeters (mm). To control and minimize the pressure of the head on the skin, a correspondingly large amount of hydrogel was applied to the area examined. After imaging the ulnar nerve in the cubital tunnel, its transverse scans were recorded in each angular position (controlled with a goniometer), always starting with full extension, then 45° angle, 90° angle, and maximum flexion of the elbow, with the measurement procedure repeated in the same order three times. The means of three measurements were used for the final analysis.

### 2.4. Statistical Analyses

Data were analyzed using STATISTICA 13.1 PL (Statsoft, Tulsa, OK, USA) software. To compare the affected with non-affected sides in different elbow positions, a repeated measurement one-way analysis of variance (ANOVA) was used. For significant main effects in the ANOVA, the planned comparisons were performed. The US results are presented in figures as mean values and 95% confidence interval (CI), whereas significant differences were presented in the text as mean differences with their 95% CI. For all analyses, the threshold of the *p*-value considered significant was set at ≤0.05.

## 3. Results

### 3.1. Characteristics of Patients with CuTS

A total of 31 patients with unilateral CuTS (10 women and 21 men) were examined. The mean age of the subjects was 54.2 (SD 8.15; range 40–68) years, mean body height was 173.7 cm (SD 11.46; range 148–189), and mean body weight was 76.16 kg (SD 14.46; range 49–102). The mean BMI was 25.24 (SD 3.89; range 20.2–44). Among the participants, 28 (90%) were right-handed and 3 (10%) were left-handed. CuTS occurred in 17 patients (55%) on the left side and in 14 (45%) on the right side. CuTS was presented in 14 patients on the dominant (right) hand and 3 patients with symptoms on the dominant (left) hand. The mean motor conduction velocity (MCV) was 35.39 m/s (SD 5.85; range 25.3–44) in the symptomatic limb and was below normal in all cases (motor conduction velocity < 49.3 m/s). The mean duration of symptoms was 14 months. The severity of clinical symptoms was from one to two according to McGowan’s classification in all patients with CuTS.

### 3.2. Shear Modulus

The ANOVA revealed significant side-to-side differences (F = 86.2, *p* < 0.001) in the ulnar nerve elasticity at 45°, 90°, and maximal elbow flexion, but not in extension (Figure 3). Compared to the non-affected side, the mean shear modulus on the affected side was higher by 52.3 kPa (95% CI 31.3–73.2), 49.5 kPa (95% CI 28.6–70.5), and 127.8 kPa (95% CI 106.9–148.7) for 45° of elbow flexion, 90° of elbow flexion, and maximal elbow flexion, respectively.

### 3.3. Cross Sectional Area (CSA)

There were significant side-to-side differences (ANOVA, F = 168.2, *p* < 0.001) in the ulnar nerve CSA value at each elbow position (extended, 45°, 90°, and maximal elbow flexion) (Figure 4). Compared to the non-affected side, the mean CSA on the affected side was higher by 6.63 mm^2^ (95% CI 4.10–5.16), 4.61 mm^2^ (95% CI 4.07–5.13), 4.19 mm^2^ (95% CI 3.66–4.72), and 4.87 mm^2^ (95% CI 4.33–5.39) for extended elbow, 45° of elbow flexion, 90° of elbow flexion, and maximal elbow flexion, respectively.

### 3.4. Trace

There were significant side-to-side differences (ANOVA, F = 54.2, *p* < 0.001) in the ulnar nerve TRACE value at each elbow position (extended, 45°, 90°, and maximal elbow flexion) (Figure 5). Compared to the non-affected side, the mean TRACE on the affected side was higher by 1.76 mm (95% CI 1.14–2.12), 1.74 mm (95% CI 1.38–2.11), 1.93 mm (95% CI 1.56–2.29), and 1.54 mm (95% CI 1.18–1.90) for extended elbow, 45° of elbow flexion, 90° of elbow flexion, and maximal elbow flexion, respectively.

## 4. Discussion

The aim of this study was to evaluate the shear modulus, CSA, and TRACE of the ulnar nerve in the cubital tunnel in different angular positions of the elbow joint in patients with CuTS. To the best of our knowledge, no studies to date have evaluated whether and how shear modulus, CSA, and TRACE values of the ulnar nerve in the cubital tunnel change in CuTS patients, depending on the angle of elbow joint. The results showed significant differences in shear modulus, CSA, and TRACE of the ulnar nerve in the symptomatic side, compared to the asymptomatic extremity. On the symptomatic side, a significantly higher ulnar nerve shear modulus was recorded in the cubital tunnel at 45°, 90°, and in full elbow flexion, compared to the asymptomatic side. Only in full extension were no significant differences found. The CSA and TRACE of the ulnar nerve in the cubital tunnel at each angular position were also greater in the symptomatic limb than in the asymptomatic limb.

SWE is a relatively new imaging technology for quantifying tissue stiffness, and its use in clinical practice is constantly being expanded [22]. SWE is used in the diagnosis of peripheral nerve neuropathies [23,24,25,26,27,28], including entrapment syndromes [23,24,25,26,27]. The most common use of SWE has been in the diagnosis of carpal tunnel syndrome [23,24,25,26]. To date, only two papers have evaluated the change in the shear modulus of the ulnar nerve in patients with CuTS [18,19]. Paluch et al. [18] showed significantly greater ulnar nerve stiffness in the cubital tunnel in CuTS patients, compared to healthy subjects. Kim and Lee, on the other hand, indicated significantly greater ulnar nerve stiffness in the cubital tunnel in CuTS patients, compared to those with medial and lateral epicondylitis [19]. In both studies, authors also assessed ulnar nerve stiffness at other levels (distal arm and mid-arm [18], and distal arm and proximal forearm [19]), but these measurements showed no significant differences, compared to asymptomatic subjects. Therefore, we decided to study only the ulnar nerve in the cubital tunnel but also in different angular positions of the elbow joint.

Our findings on shear modulus of the ulnar nerve in the cubital tunnel in CuTS patients are consistent with those reported in previous studies [18,19] and show that nerve stiffness is significantly higher on the symptomatic side in CuTS patients. Paluch et al. [18] examined the ulnar nerve in the flexion of the elbow joint but did not accurately determine the angle of flexion. Kim and Lee [19] studied the ulnar nerve at 30° of flexion. The mean SWE values obtained at the level of the cubital tunnel in patients with CuTS by Paluch et al. [18] were 96.38 kPa, whereas data reported by Kim and Lee [19] were 66.8 kPa. The difference between patients with CuTS and a control group of healthy people in the study by Paluch et al. [18] was 63.3 kPa, whereas in the study by Kim and Lee [19], the difference between patients with CuTS and those with epicondylitis was 45.6 kPa. In our study, the mean difference was 52.3 kPa at 45° of flexion, 49.5 kPa at 90°, and as much as 127.8 kPa at the maximum elbow joint flexion. These results indicate that nerve stiffness increases significantly in the maximum elbow joint flexion and that it is important to control the degree of elbow joint flexion during SWE examination of the ulnar nerve in the cubital tunnel, as this has a significant effect on nerve stiffness. Paluch et al. [18] emphasized the lack of standardization of ultrasonography of peripheral nerves, so it seems that the standardization of SWE of the ulnar nerve in the cubital tunnel should take into account the angle of flexion of the elbow joint.

The usefulness of CSA assessment in the diagnosis of various peripheral neuropathies, including CuTS, has been much more frequently demonstrated in scientific studies [14,15,16,29]. However, similar to the assessment of SWE of the ulnar nerve, the effect of the degree of elbow joint flexion on CSA in CuTS patients has not been studied. Previous studies have shown that normal CSA of the ulnar nerve in the cubital tunnel range from 6 to 9 mm^2^, and the cutoff value suggestive of CuTS is 10 mm^2^ [14]. Furthermore, previous studies have mostly evaluated CSA in full extension and 90° of flexion [30,31], in 30° of flexion [32], and in 90° of elbow joint flexion only [22,33]. In our study, the mean CSA of the ulnar nerve at the level of the cubital tunnel in the symptomatic limb was 10.9 mm^2^, while in the asymptomatic limb, this was 6.48 mm^2^, which is consistent with previous findings [14]. The significant difference in CSA between symptomatic and asymptomatic limbs confirms the validity of using the asymptomatic limb as a reference limb. Such observations have also been made by previous authors [24]. In our study, the largest CSA difference between the symptomatic and asymptomatic extremities occurred at full extension, whereas CSA was similar for subsequent angular positions. We found no significant change in CSA at different angles of elbow joint flexion, again consistent with previous studies [23,24]. Therefore, it can be concluded that unlike shear modulus assessment, where nerve stiffness increases with the degree of elbow joint flexion, such a relationship does not apply to CSA. Thus, in the case of CSA, its greatest diagnostic value is its significantly higher value on the symptomatic side versus the asymptomatic side. Therefore, in the case of CSA of the ulnar nerve in the cubital tunnel, the comparison with the reference limb and the cutoff values are important in the diagnosis of CuTS, and the elbow flexion angle has less effect.

Our study also showed that the TRACE value on the symptomatic side of the elbow joint was significantly longer on each angular position, compared to the asymptomatic limb (extension: 17.6 mm; 45° of flexion: 17.4 mm; 90° of flexion: 1.93 mm: maximum flexion: 15.4 mm). No studies in the available literature have explored how TRACE changes in patients with CuTS in the symptomatic, compared to the asymptomatic, limb and it is difficult to refer to other studies in this case. Nevertheless, the significant differences found between the symptomatic and asymptomatic limbs in CuTS patients indicate that TRACE may also have diagnostic potential in this neuropathy. Further studies may also determine a cutoff value for TRACE, as is the case for CSA. However, TRACE seems to be more prone to measurement errors than CSA, as the circumference of the whole nerve over the inner hypoechoic border has to be manually drawn.

The increased stiffness of the ulnar nerve in CuTS patients may be due to its fibrosis caused by chronic compression. Chronic compression of the nerve impairs blood supply and microcirculation in the nerve, leading to capillary leakage and swelling of the nerve, and, over time, the development of chronic inflammation, demyelination, and fibrosis [18,34]. Increased nerve stiffness can also result from the compression of the nerve by surrounding tissues [19]. This, in turn, affects the increase in endoneural pressure, which may result in increased stiffness. This could explain the gradual increase in ulnar nerve stiffness in the cubital tunnel with increasing degrees of elbow joint flexion that we observed in our study. This can also be explained by the decrease in the lumen of the cubital tunnel with the angle of flexion of the elbow joint, and thus the increase in pressure in the cubital tunnel [20]. It is also likely that during the flexion of the elbow joint, the stiffness of the ulnar nerve increases due to its stretching, which is greatest at the level of the cubital tunnel and increases by 23% under physiological conditions [21]. Further, if the compression of the nerve in CuTS limits its mobility, it can be stretched even more at this level. It is also likely that all of these intrinsic and extrinsic mechanisms lead to increased nerve stiffness. Increased CSA of the ulnar nerve in CuTS is explained by swelling, usually proximal to the compression site, loss of normal nerve bundle patterns, and decreased nerve mobility [12]. Furthermore, increasing CSA leads to the lengthening of the nerve trace (TRACE value). It should be emphasized that most studies have shown that the site of the greatest nerve swelling is the cubital tunnel at the level of the medial epicondyle of the humerus [14].

Our study has some limitations. The first limitation is the relatively small group of CuTS patients studied. The second limitation is that nerve conduction examinations were only performed in the symptomatic limb and not in the asymptomatic limb. Consequently, the presence of nerve conduction disorders in the non-symptomatic extremity cannot be ruled out even before the symptoms. On the other hand, the subjects did not report any subjective symptoms of ulnar nerve neuropathy and side-to-side differences were clearly observed. Such a study would also be unethical because of the burdensome nature of electrodiagnostic findings. The third limitation is the lack of verification of other pathologies, which may be a cause of ulnar nerve compression in elbow region. It is worth noting that there are many potential causes of CuTS, but they are often difficult to identify. Mezian et al. [12] have suggested a number of potential causes of CuTS (ganglion, heterotopic ossification, anconeus epitrochlearis accessory muscle, peripheral nerve tumors, etc.). Some other authors have considered nerve instability as a cause of CuTS [35]. Calfee et al. [36] have indicated that ulnar nerve hypermobility may occur between 2% and 47% of asymptomatic individuals. Van Den Berg et al. [37] have reported the occurrence of ulnar nerve subluxation and dislocation in 5.7% of subjects. All of these aspects were not verified in this study and could have affected US measurements. However, we did not find any signs of instability or hypermobility of the ulnar nerve in any of the examined patients during the measurements (patients were examined in various elbow positions).

## 5. Conclusions

The ulnar nerve in patients with CuTS is characterized by greater stiffness, CSA, and TRACE values in the cubital tunnel on the symptomatic, compared to the asymptomatic, side in different angular positions of the elbow joint. Therefore, US examination (shear modulus, CSA, and TRACE) of the ulnar nerve can aid in the diagnosis and monitoring of lesions in patients with CuTS. Further research in this area is warranted and necessary.

## Figures and Tables

**Figure 1 sensors-22-08354-f001:**
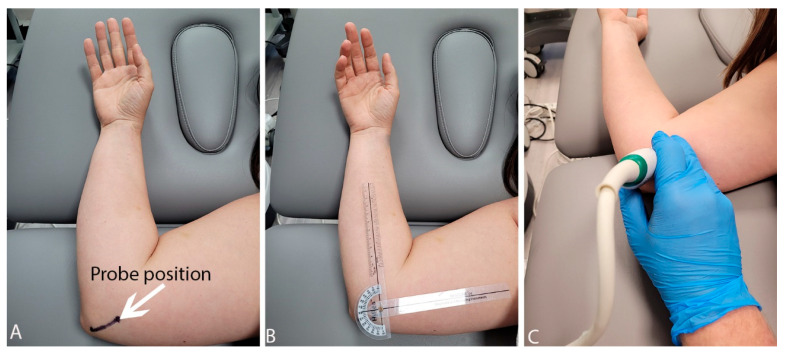
Probe position (**A**), control of the position of the elbow joint (**B**) and examination phase (**C**).

**Figure 2 sensors-22-08354-f002:**
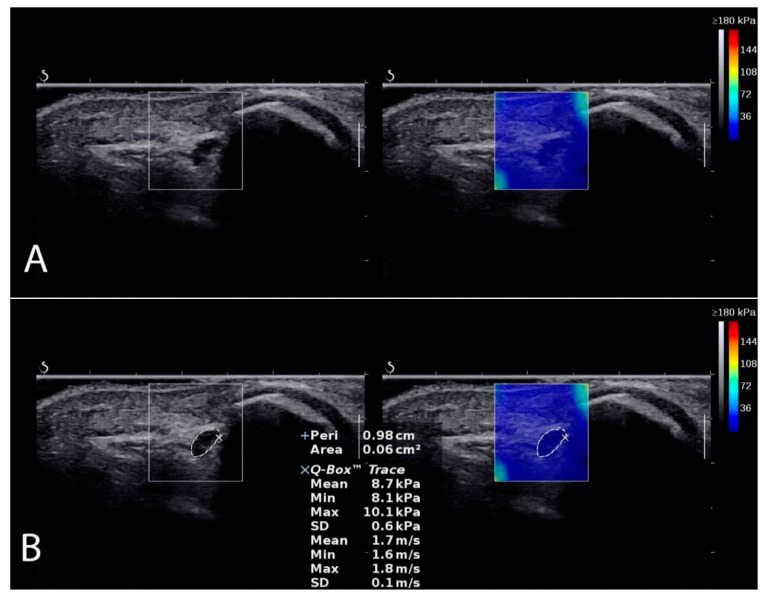
Ulnar nerve localization in SWE mode (**A**). Measurement example (**B**) of CSA, shear modulus, and TRACE.

**Figure 3 sensors-22-08354-f003:**
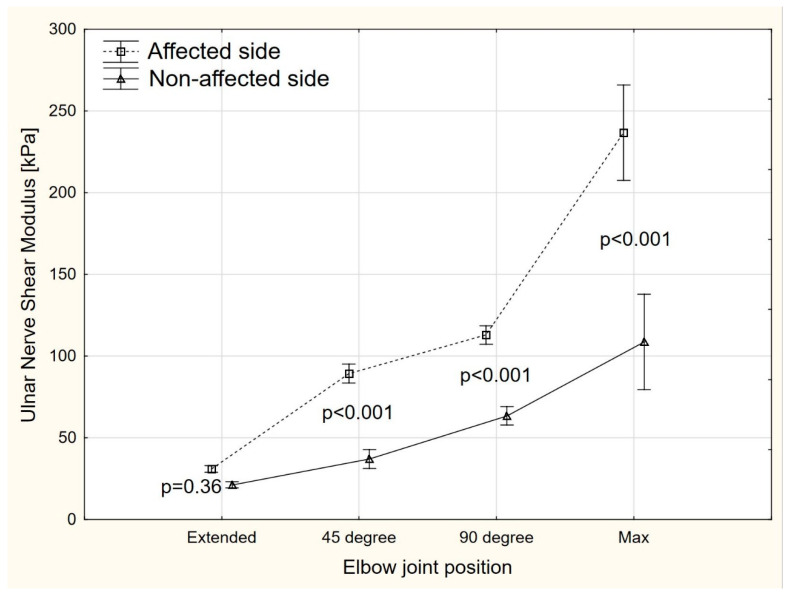
The share modulus changes in different positions of the elbow joint.

**Figure 4 sensors-22-08354-f004:**
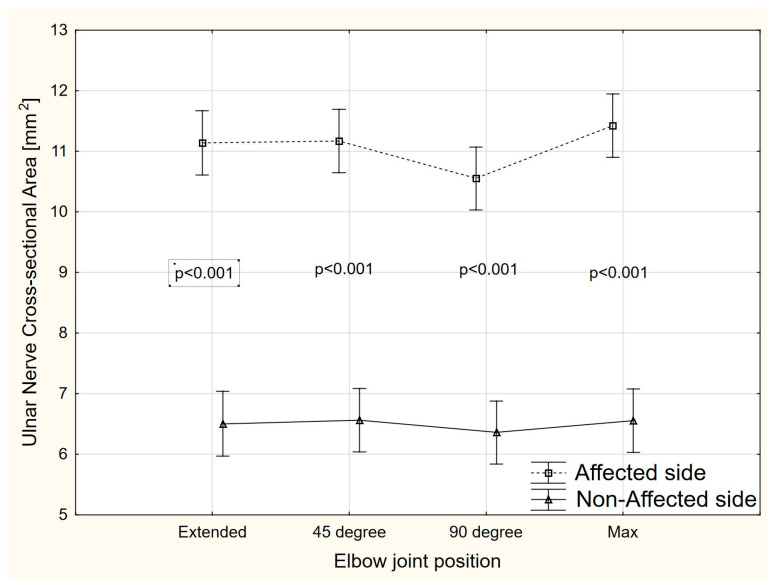
The CSA changes in different positions of the elbow joint.

**Figure 5 sensors-22-08354-f005:**
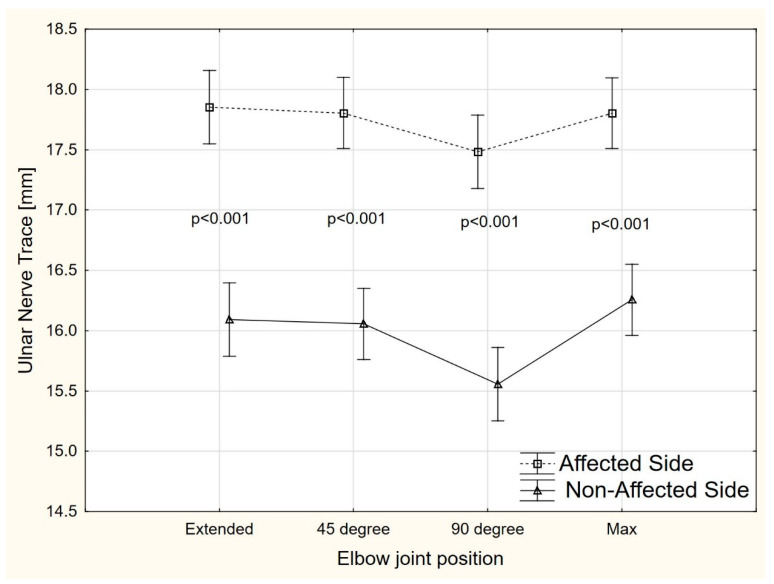
The TRACE changes in different positions of the elbow joint.

## Data Availability

Data are available on request.

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
