# Peer review of "Changes in Ultrasound Measurements of the Ulnar Nerve at Different Elbow Joint Positions in Patients with Cubital Tunnel Syndrome"

_sensors, 2022, doi:10.3390/s22218354_

Round 1
Reviewer 1 Report
I enjoyed reading this well-written article. The research question is highly pertinent to clinicians using diagnostic ultrasound and taking care of ulnar neuropathy patients. The study is well designed and well standardized (good job on having a blinded US examiner).
Here are my other comments & suggestions.
Intro:
Typo: authors; authors’: To the best of the authors’ knowledge
Results
In addition to that sentence “CuTS occurred in 17 patients 145 (55%) on the left side and in 14 (45%) on the right side.” It would be nice to know how many patients were symptomatic in their DOMINANT limb.
Also, it would be VERY INTERESTING to have data on DURATION OF SYMPTOMS and potentially a correlation between duration of symptoms and CSA / SWE (more persistent symptoms, stiffer nerve?) although this falls slightly outside the research question.
Discussion:
The sentence (Our study also showed that the TRACE value on the symptomatic side of the elbow 250 joint was significantly longer on each angular position compared to the asymptomatic 251 limb) should be clarified so that it is made clear that the values presented are the “delta” / difference between TRACE measures in symptomatic—asymptomatic limbs and NOT absolute TRACE values.
typo 174mm: 1.74mm (45° of flexion: 174 mm)
How does TRACE correlate to CSA? If the correlation is near perfect, either both, or the one less prone to measurement errors should be used.
“However, TRACE may be found to be more prone to measurement errors.” Is there a reference for that statement? If not, please explain in more detail for the readers.
Were anatomical variations predisposing to cubital tunnel syndrome (e.g. anconeus epitrochlearis) taken into account on US examination?
Were patients evaluated for ulnar nerve instability in elbow flexion? This is important and should be discussed in the article, even though the association between ulnar nerve hypermobility and ulnar neuropathy seems controversial. (https://link.springer.com/article/10.1007/s40477-019-00370-9
https://www.sciencedirect.com/science/article/pii/S0003999321002318?casa_token=WgV11GZQPm8AAAAA:CVdKz-cJBDcJGMqdk9Y8DpKtz2-Ne3lBHM3WsFJhv2CmY_UbcBihKtnlj_-N2qY48q9v29vaqew8#
https://pubmed.ncbi.nlm.nih.gov/21123610/
https://journals.lww.com/ajpmr/Fulltext/2022/06000/EURO_MUSCULUS_USPRM_Dynamic_Ultrasound_Protocols.14.aspx?casa_token=202kRwDLUrsAAAAA:rZzWjzmBlAZH3FjSkU4Cm-USV4pItJN_o_QTEkFPApoa7ASzP8HkAq-iVMsw7lfNgFuN9hCltB0-36k5PhsNI08WwVmw).
Other than the fact that ulnar nerve hypermobility might not be related to the risk of developing ulnar neuropathy at the elbow, it might influence SWE measurements and should therefore be reported in the article, as it is a common condition (37% https://journals.lww.com/ajpmr/Fulltext/2022/06000/EURO_MUSCULUS_USPRM_Dynamic_Ultrasound_Protocols.14.aspx?casa_token=202kRwDLUrsAAAAA:rZzWjzmBlAZH3FjSkU4Cm-USV4pItJN_o_QTEkFPApoa7ASzP8HkAq-iVMsw7lfNgFuN9hCltB0-36k5PhsNI08WwVmw / 56 % https://link.springer.com/article/10.1007/s40477-019-00370-9 ).
Author Response
I enjoyed reading this well-written article. The research question is highly pertinent to clinicians using diagnostic ultrasound and taking care of ulnar neuropathy patients. The study is well designed and well standardized (good job on having a blinded US examiner).
Response: We appreciate the time and effort that you have dedicated to review our manuscript. We would also like to thank you for all your comments. We hope that the present form of the paper meets your expectation.
Here are my other comments & suggestions.
Intro: Typo: authors; authors’: To the best of the authors’ knowledge
Response: Thanks for your suggestion. Typo has been corrected.
Results In addition to that sentence “CuTS occurred in 17 patients 145 (55%) on the left side and in 14 (45%) on the right side.” It would be nice to know how many patients were symptomatic in their DOMINANT limb.
Response: Thanks for your suggestion. We added the following sentence: “CuTS was presented in 14 patients on the dominant (right) hand and 3 patients with symptoms one the dominant (left) hand”.
Also, it would be VERY INTERESTING to have data on DURATION OF SYMPTOMS and potentially a correlation between duration of symptoms and CSA / SWE (more persistent symptoms, stiffer nerve?) although this falls slightly outside the research question.
Response: Thanks for your suggestion. We added the following sentence:”The mean duration of symptoms was 14 months. The severity of clinical symptoms was from 1 to 2 according to McGowan's classification in all patients with CuTS.”
Discussion: The sentence (Our study also showed that the TRACE value on the symptomatic side of the elbow 250 joint was significantly longer on each angular position compared to the asymptomatic 251 limb) should be clarified so that it is made clear that the values presented are the “delta” / difference between TRACE measures in symptomatic—asymptomatic limbs and NOT absolute TRACE values. typo 174mm: 1.74mm (45° of flexion: 174 mm)
Response: Thank you for your valuable attention. All are absolute values. The dot was wrongly placed
How does TRACE correlate to CSA? If the correlation is near perfect, either both, or the one less prone to measurement errors should be used.
Response: We did not investigate the correlation between TRACE and CSA as this was not the purpose of our study. Thus, we have decided to place both of them.
“However, TRACE may be found to be more prone to measurement errors.” Is there a reference for that statement? If not, please explain in more detail for the readers.
Response: There are no studies analyzing which measurement (Trace or CSA) is more prone to measurement error. Thus, we have rewritten the sentence as follows:
However, TRACE seems to be more prone to measurement errors than CSA, as circumference of the whole nerve over the inner hypoechoic border has to be manually draw.
Were anatomical variations predisposing to cubital tunnel syndrome (e.g. anconeus epitrochlearis) taken into account on US examination?
Response: Researchers indicate many pathologies within the elbow that may be the cause of potential nerve compression (ganglion, heterotopic ossification, anconeus epitrochlearis accessory muscle, peripheral nerve tumours, etc.) Thus, it would be difficult to identify and take into account all of them. This was not the purpose of our work, but we decided to mentioned in on the limitation of our study.
We have added the following sentence: “The third limitation is the lack of verification other pathologies which may cause of potential ulnar nerve compression in elbow region.”
Were patients evaluated for ulnar nerve instability in elbow flexion? This is important and should be discussed in the article, even though the association between ulnar nerve hypermobility and ulnar neuropathy seems controversial. (https://link.springer.com/article/10.1007/s40477-019-00370-9
https://www.sciencedirect.com/science/article/pii/S0003999321002318?casa_token=WgV11GZQPm8AAAAA:CVdKz-cJBDcJGMqdk9Y8DpKtz2-Ne3lBHM3WsFJhv2CmY_UbcBihKtnlj_-N2qY48q9v29vaqew8#
https://pubmed.ncbi.nlm.nih.gov/21123610/
https://journals.lww.com/ajpmr/Fulltext/2022/06000/EURO_MUSCULUS_USPRM_Dynamic_Ultrasound_Protocols.14.aspx?casa_token=202kRwDLUrsAAAAA:rZzWjzmBlAZH3FjSkU4Cm-USV4pItJN_o_QTEkFPApoa7ASzP8HkAq-iVMsw7lfNgFuN9hCltB0-36k5PhsNI08WwVmw).
Other than the fact that ulnar nerve hypermobility might not be related to the risk of developing ulnar neuropathy at the elbow, it might influence SWE measurements and should therefore be reported in the article, as it is a common condition (37% https://journals.lww.com/ajpmr/Fulltext/2022/06000/EURO_MUSCULUS_USPRM_Dynamic_Ultrasound_Protocols.14.aspx?casa_token=202kRwDLUrsAAAAA:rZzWjzmBlAZH3FjSkU4Cm-USV4pItJN_o_QTEkFPApoa7ASzP8HkAq-iVMsw7lfNgFuN9hCltB0-36k5PhsNI08WwVmw / 56 % https://link.springer.com/article/10.1007/s40477-019-00370-9 ).
Response: Thanks for your suggestion. We did not find any instability or hypermobility of the ulnar nerve in any of the examined patients. We examined patients in various angular positions (including full flexion) of the elbow joint, also in a dynamic examination to observe the behavior of the ulnar nerve. Accordingly, we have added the following info:
The third limitation is the lack of verification other pathologies which may be a cause of ulnar nerve compression in elbow region. It is worth noting that there are many potential causes of CuTS, but they are often difficult to identify. Mezian et al. [12] have suggested a number of potential cause of CuTS (ganglion, heterotopic ossification, anconeus epitrochlearis accessory muscle, peripheral nerve tumors, etc.). Some other authors have considered nerve instability as a cause of CuTS [36]. Calfee et al. [37] have indicated that ulnar nerve hypermobility may occur between 2% and 47% of asymptomatic individuals. Van Den Berg et al. [38] have reported the occurrence of ulnar nerve subluxation and dislocation in 5.7% of subjects. All of these aspects were not verified in this study and may affected US measurements. However, we did not find any signs of instability or hypermobility of the ulnar nerve in any of the examined patients during the measurements (patients were examined in various elbow positions).
Reviewer 2 Report
Hi, unfortuaetly your paper is insufficient and contains no new information to the field.
Author Response
Response: appreciate the time and effort that you have dedicated to review our manuscript, but we respectfully disagree with the reviewer about this report. To the best of our knowledge there is no studies analysing ulnar nerve morphology (especially shear modulus) in relation to CuTS taking into account different elbow position. If the information in the article is so obvious (no new information), please provide the studies reporting such results. From our perspective, we do not understand this type of opinion from a reviewer without providing any objective info.
Reviewer 3 Report
This paper deals with the sonographic diagnosis of a cubital tunnel syndrome. The results may have impact on clinical practice.
- Methods 2.1: it should be explained what a „physiotherapeut“ is, - because in different countries it has different meanings
- why did the authors use only a 10 MHz probe? Since the region of the nerve is superficial, higher MHz could be used for better resolution
- 2.3 Ultrasound Measurement: the different measures should be described more in detail, especially TRACE
- in the conclusion the authors should outline the meaning of the different angles - what is the best angle for the investigation?
Author Response
This paper deals with the sonographic diagnosis of a cubital tunnel syndrome. The results may have impact on clinical practice.
Response: We appreciate the time and effort that you have dedicated to review our manuscript. Thank you for appreciating our work and its clinical importance for clinicians using ultrasound diagnostics in ulnar neuropathy. We would like to thank you for your suggestions. The manuscript has been revised as recommended to make it better for the readers.
- Methods 2.1: it should be explained what a „physiotherapeut“ is, - because in different countries it has different meanings
Response: We obviously referred to physiotherapist, a healthcare profession worldwide recognized a someone who treats people using physiotherapy (Definition of physiotherapist from the Cambridge Advanced Learner's Dictionary & Thesaurus © Cambridge University Press). In American English this is a physical therapist.
why did the authors use only a 10 MHz probe? Since the region of the nerve is superficial, higher MHz could be used for better resolution
Response: Of course, we agree that the higher the frequency, the better the image quality, especially in superficial tissues. Our ultrasound scanner was only coupled with a linear transducer array (2-10 MHz; SuperLinear 10-2, Vermon, Tours, France), and therefore such a transducer was used by us in our research.
- 2.3 Ultrasound Measurement: the different measures should be described more in detail, especially TRACE
Response: Added: The CSA is the cross-section of the ulnar nerve in the ulnar tunnel measured in square millimeters (mm2). The share modulus evaluated ulnar nerve stiffness in kilopascals (kPa). The TRACE, in turn, is the circumference of the ulnar nerve over the inner hypoechoic border of the ulnar nerve measured in milimeters (mm).
- in the conclusion the authors should outline the meaning of the different angles - what is the best angle for the investigation?
Response: The results obtained by us do not allow us to say which angle of flexion is the best for ultrasound measurements of the elbow joint. Our research only shows how the ultrasound parameters we measure (share modulus, CSA and TRACE) change depending on the elbow flexion angle.
Round 2
Reviewer 2 Report
Dear Authors,
I dont see any new information in your manuscript. I therefore cannot recommend to publish your article.